# Degradation of Hydraulic System due to Wear Particles or Medium Test Dust

Nejc Novak, Ana Trajkovski, Mitjan Kalin and Franc Majdič *

Faculty of Mechanical Engineering, University of Ljubljana, 1000 Ljubljana, Slovenia;
nejc.novak@fs.uni-lj.si (N.N.); ana.trajkovski@fs.uni-lj.si (A.T.); mitjan.kalin@fs.uni-lj.si (M.K.)
* Correspondence: franc.majdic@fs.uni-lj.si; Tel.: +386-1-47-71-411

**Abstract:** Contamination in hydraulic systems is the cause of 70% of failures. This study highlights the performance degradation caused by solid particle contamination of hydraulic components: hydraulic gear pump, 4/3 valve, and orbital motor. Experimental durability tests of components with wear particles and test dust are used to investigate the effects of accelerated wear caused by these two types of contaminants. Results show that oil contaminated with wear particles reduces the volumetric efficiency of the gear pump by 18% and the hydraulic valve by only 0.8%, while oil contaminated with test dust reduces the efficiency of the pump by 76% and the hydraulic valve by 0.9%. This research provides insights for accelerating hydraulic component testing to improve system reliability and longevity.

**Keywords:** hydraulic system; oil cleanliness; gear pump; directional 4/3 valve; orbital motor; wear; internal leakage; volumetric efficiency

## 1. Introduction

Hydraulic system life refers to the period of time during which a machine can continuously perform its intended function under specified temperatures, pressures, and flow rates. Overall efficiency and service life are mainly influenced by factors such as hydraulic oil quality, temperature, and oil cleanliness. More than 70% of failures are due to the contaminant in hydraulic oil, with 60–70% of total failures due to solid particles [1]. The level of cleanliness of hydraulic fluids is one of the most important indicators for predicting possible failures of hydraulic components and consequently of the whole system [2–4]. There are several international standards for measuring the cleanliness of fluids, such as ISO 4406 [5] and SAE AS 4059 [6]. Using a fluid sample, the particle concentration and size distribution are evaluated according to the standard [5].

Thus, in the hydraulic system, there are not only wear particles but also particles that enter the system as contaminants from the environment or are present in the system from the beginning. The size and composition of the particles have a decisive influence on the system performance, especially if the gap between two mutually moving surfaces is similar in size to the particles themselves [7]. Particles entering such contacts cause wear [8] through mechanisms such as three-body abrasion [9] and erosion [10]. As a result of wear on the sliding surfaces, wear particles are generated and move through the system with the fluid [11]. The number of particles in the fluid is controlled by appropriate filtration [12]. Cleanliness stabilization is achieved when the number of particles in the hydraulic oil does not increase nor decrease but is constant. The increase in particles is due to wear of hydraulic components, built-in contaminants from manufacturing [2], and the decrease in particles is due to filtering of the oil [13]. When an equilibrium is reached between an input and output of the particles, the cleanliness stabilizes.

To improve the quality of hydraulic filters and to calibrate the particle counters, test dust was introduced [14]. According to ISO 12103-1, A3 MTD, test dust has a well-defined particle size distribution and concentration [13]. Medium test dust (MTD) is used for testing

various types of hardware devices. Since test dust is harder than typical wear particles [15], its use for acceleration testing has been explored [16]. Therefore, it is used to accelerate the wear of various hydraulic components [17–20]. The increase in wear leads to an increase in leakage flow in hydraulic components and, moreover, to a decrease in the volumetric efficiency of hydraulic components [21].

Gear pumps are representative of hydraulic components, which have already been studied by Ranganathan et al. [22], Frith [23], and Shin [20] using test dust. They studied the chemical composition, size distribution, and concentration of test dust on the wear of the pump and consequently the loss of an effective flow rate. Studies on various valves, e.g., ball-seat valves [24] and directional valves [25], showed that performance decreased with increasing internal leakage. It was found that most of the system performance degradation occurs in the phase with improper filtration [25]. Zhang et al. [26] showed that the wear of the basic moving elements of the hydraulic motor leads to an increase in leakage. Corrosion wear can also enhance leakage. The corrosion of surfaces is a chemical process that converts metal surface atoms into a more chemically stable oxide. Pieces of debris are newly formed chemical compounds, usually agglomerated or sometimes mixed with fragments of surface materials [27].

Particles damage each component in the hydraulic system, so wear occurs. The wear of the sealing surfaces in the hydraulic system is noticeable by leakage and consequently by a loss in volumetric efficiency of the whole system. A review of the literature shows that while test dust accelerates wear compared to wear particles typically present in hydraulic system fluid, it is still a challenge to determine the acceleration factor. There is no direct comparison in the literature of the effect of wear particles and test dust on the wear performance of hydraulic components. In our laboratory, two durability tests of hydraulic systems were carried out: one with wear particles and the other with test dust. The study presents the design and test rig itself, the calculation of the wear parameter of the pump, and the efficiency of the pump, 4/3 directional valve, and hydraulic motor over time.

## 2. Materials and Methods

### 2.1. Test Rig

Two identical hydraulic test rigs were set up in our laboratory. The effect of oil cleanliness in the hydraulic system on the durability of the system itself was tested, and the effect of wear particles or test dust (MTD) was compared. Figure 1 shows both test rigs, with wear particles added to the first test rig (right) and test dust added to the second test rig (left). Initially, 30 L of hydraulic oil ISO VG 46 was added to both units. Flow through the 2SP A06 D 10-G gear pump was measured at a displacement of 6 cm$^3$/rev and a pressure of up to 270 bar. Internal leakage was measured on a 4/3 directional control valve with a solenoid model DS3-S1/11-N-D24K1. In the zero position of the directional control valve, all ports were closed (P, T, A, B). The oil temperature in the tank was about 65 °C. The thermostat, which switched the oil cooling on or off, was set to 60 °C. The directional control valve was switched with a frequency of 5 Hz.

The test rigs developed in the Laboratory for Hydraulic Power and Controls consisted of all the key hydraulic components that are also commonly used in the field. Table 1 shows the bill of material (BOM) of the test rigs with the corresponding system positions.

The pump (pos. 1, Figure 2) sucked the oil from the tank (pos. 19, Figure 2), which was inclined at an angle of 15° (to prevent particles from settling, the suction pipe was located at the lowest point (corner) of the tank), and pushed the oil through the check valve (pos. 12, Figure 2) to directional 4/3 valve with solenoid (pos. 4, Figure 2). During the parallel or diagonal positions of the valve, the oil flowed through working lines A or B, the orifices (pos. 15, Figure 2), hydraulic motor (pos. 6, Figure 2), manually operated 3/2 valve (pos. 17, Figure 2), cooler (pos. 9, Figure 2), priority valve (pos. 7, Figure 2), filter (pos. 8, Figure 2), and back to the reservoir (pos. 19, Figure 2). The priority valve can be set to an adjustable pressure differential so that it can divert the oil flow around the filter into reservoir. In this way, the oil can be filtered and the cleanliness level is consequently

higher, or the oil bypasses the filter and the cleanliness level decreases. Factory values of the maximum permissible pressure of all tested hydraulic components were inspected, and the pressure was set to somewhat lower at 200 bar. The hydraulic oil in both test systems was initially filtered to a cleanliness level of 16/15/13 or less. The temperature was monitored and ranged between 60 °C and 70 °C during the test. Cleanliness was monitored by manually controlling the 3/2 valve to position 17 and then taking oil samples for cleanliness measurements through the test port on the pressure relief valve. Control spool inside a directional 4/3 valve had the function of redirecting fluid flow (fluid energy). It made a connection from port P (pressure port of the pump) to port A (working port) and at the same time connection from port B (another working port) to port T (port of the tank). Described position is called parallel position. The diagonal position connected port P to port B and port A to port T; in this case, the fluid flow was reversed from the directional valve onwards. This is necessary when actuating hydraulic cylinders for extending and retracting piston and rod in and out of the housing of the cylinder or rotating hydraulic motor in desired direction (clockwise, counter clockwise). The duration of one cycle was 0.2 s, i.e., solenoid a was activated so that the spool of the directional 4/3 valve reached a parallel position, and then solenoid b was activated so that the spool returned to its initial position, i.e., the diagonal position. The pressure conditions in the system using pressure measurement system (Service Master Plus K-SCM-500-01-01, Parker Hannfin, Bielefeld, Germany) and pressure sensor (model A-10, Wika, Klingenberg, Germany) 0–400 bar with 0–10 V output were measured. An air oil cooler with a thermal switch (located in the reservoir) that determined the temperature of the reservoir was used. The orifice diameter was determined by the system pressure and flow rate. Using these values and Bernoulli's equation, the orifice diameter was determined.

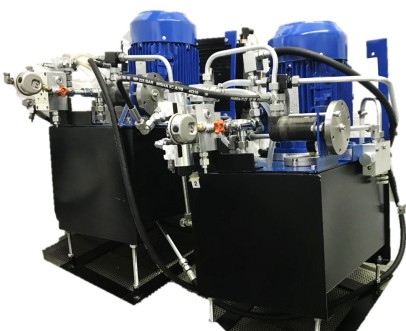

**Figure 1.** Two identical hydraulic system durability test rigs with wear particles (right, front) and test dust (MTD) (left, rear).

**Table 1.** BOM for the hydraulic scheme for the durability test of the test rig.

| Position | Item Description |
|---|---|
| 1 | Gear pump with fix displacement |
| 2 | Electric motor |
| 3 | Pressure relief valve |
| 4 | 4/3 directional valve with solenoid |
| 5 | Hydraulic block |
| 6 | Hydraulic motor |
| 7 | Hydraulic priority valve |
| 8 | Filter |
| 9 | Cooler |
| 10 | Electric motor for vent |
| 11 | Pressure relief valve for measuring |
| 12 | Check valve |
| 13 | Manometer |
| 14 | Pressure sensor |
| 15 | Orifice |
| 16 | Temperature and oil level switch |
| 17 | Manually operated 3/2 valve |
| 18 | Flexible hydraulic hose |
| 19 | Reservoir |
| 20 | Pressure relief valve for measuring |
| 21 | Quick coupling for measurement |

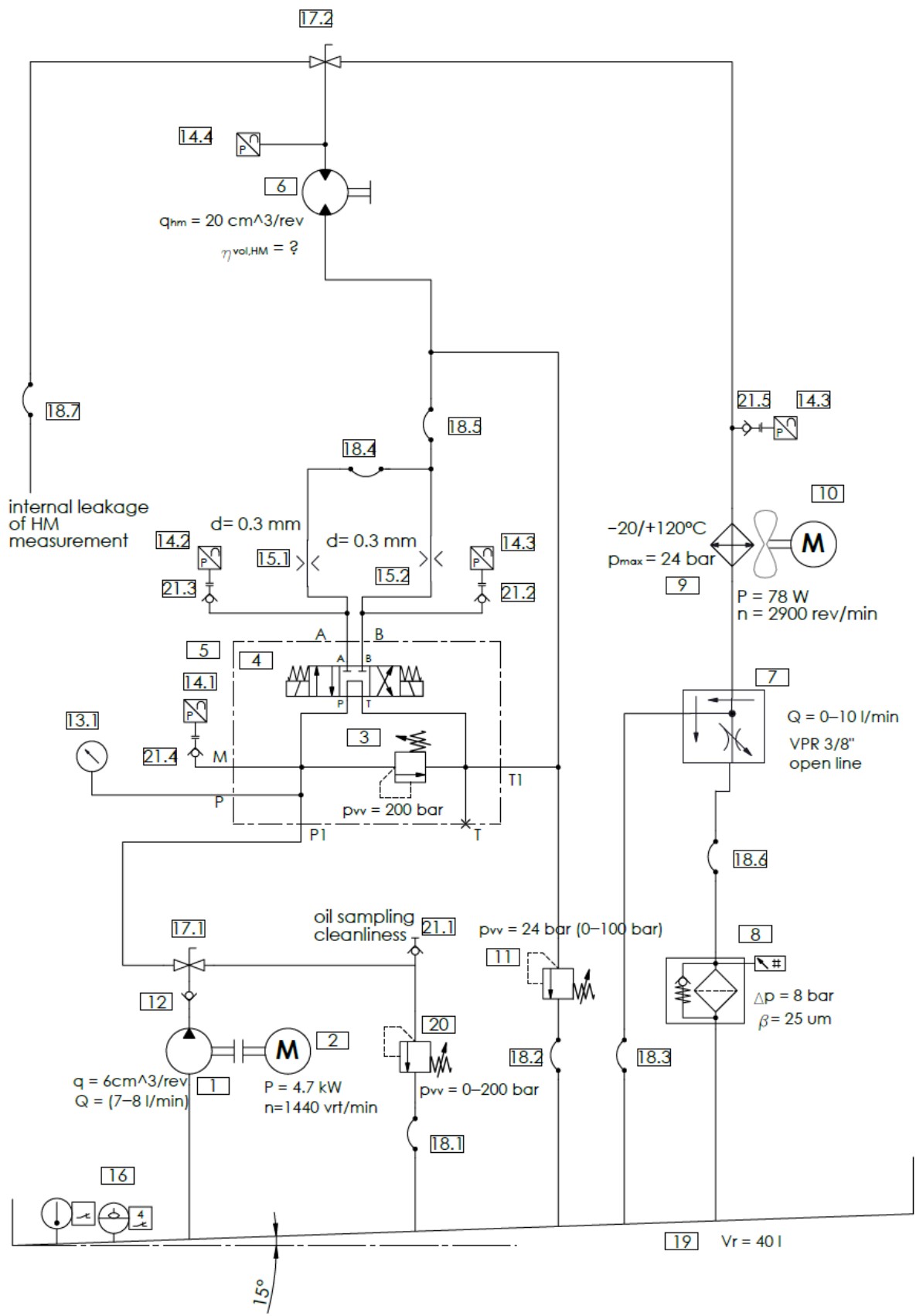

**Figure 2.** Hydraulic scheme for durability test of hydraulic gear pump.

At the beginning of the test, 5 g of wear particles was put in the first test rig and 5 g of test dust (MTD) in the second test rig. Then, 1 g of contaminant to ensure oil cleanliness 22/21/20 was added. There are several ways to measure oil cleanliness. The particle counter (iCount LaserCM, Parker Hannifin, Hydraulic Filter Division, Europe) was used to measure the cleanliness together with the bottle sampler. First, 2 dL of oil samples was taken from the test rig and air bubbles were vacuumed out to reduce the measurement error. Then, the bottle sampler was flushed with its function to ensure that only the oil present was measured. Then, the cleanliness of the same sample was measured 3 times.

### 2.2. Hydraulic External Gear Pump—Specimen 1

The volumetric efficiency of the gear pump was measured with an additional pressure relief valve by switching the manually operated 3/2 ball valve (pos. 17.1, Figure 2), which returned the oil to the tank through the pressure relief valve (pos. 20, Figure 2). Measurements were made at four different pressures: 0 bar, 100 bar, 150 bar, and 200 bar. Then, flow rates were measured with a stopwatch and a weight scale, weighing the oil volume in graduated cylinder. From the density of the oil and the weight of the oil, the volume of the oil and thus the flow rate, according to Equation (1), were indirectly determined:

$$Q = \frac{V}{t} \tag{1}$$

where $Q$ is the flow rate in L/min, $V$ is the volume of oil m$^3$, and $t$ is the time in min. The gear pump model 2SP A06 D 10-G consisted of the front cover, the side end plate, the main and auxiliary gear shafts, the housing, the second side plate, and the rear cover (Figure 3). Leakage occurred between the gears (path b, Figure 3) and between the gear and the housing (path a, Figure 3). Pressure of the fluid is increasing in steps between the suction and pressurized side (where gear and housing are containing fluid) because of the construction of the pump.

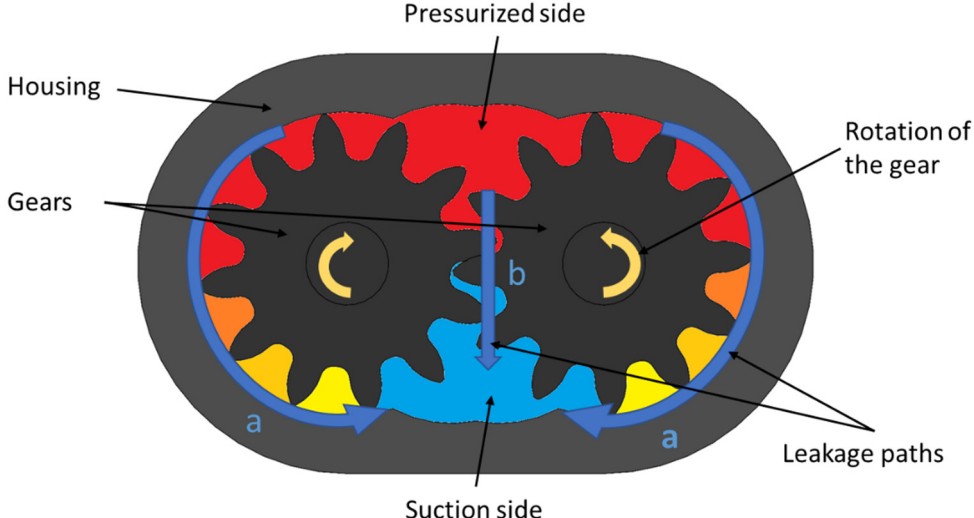

**Figure 3.** Schematic representation of leakage paths: (a) between the gear and housing and (b) between the two gears.

The actual volumetric pump efficiency ($\eta_{V,act}$), according to equation Equation (2), was in correlation with the pump flow rate.

$$\eta_{V,act} = \eta_{V,theor} \frac{n_{act}}{n_{theor}} \tag{2}$$

where $\eta_{V,theor}$ is the theoretical volumetric efficiency calculated as the maximum flow rate (at atmospheric pressure) of the pump divided by the actual flow rate (at actual pressure),

$n_{act}$ is the actual rotational speed, and $n_{theor}$ is the theoretical rotational speed in rpm. When calculating the flow rate of the pump, it is usually assumed that the rotational speed of the electric motor shaft is constant at all applied pressures, but in reality, it decreases with increasing pressure at the outlet of the pump (torque increases). Also, as the rotational speed increases, the volumetric efficiency increases due to the hydrodynamics inside the pump and then forms a curve with the highest volumetric efficiency at certain speed. If the rotational speed continues to increase, the volumetric efficiency decreases. Table 2 lists the measured rotational speeds at various pressure points. The wearable surfaces on the pump housing are shown in Figure 4. This wear occurred mostly on the suction side of the pump, but was also observed on the pressurized side. The housing was cut open and examined with digital microscope (Hirox HRX-01 and NPS, Hirox Europe, Limonest, France). Examined surfaces are showed on Figure 4d.

**Table 2.** Difference between actual rotational speed and theoretical rotational speed.

| Pressure p, Bar | Actual Rotational Speed $n_{act}$, rev/min | Theoretical Rotational Speed $n_{theor}$, rev/min |
|:---:|:---:|:---:|
| 0 | 1498 | 1500 |
| 50 | 1489 | 1500 |
| 100 | 1480 | 1500 |
| 150 | 1465 | 1500 |
| 200 | 1460 | 1500 |

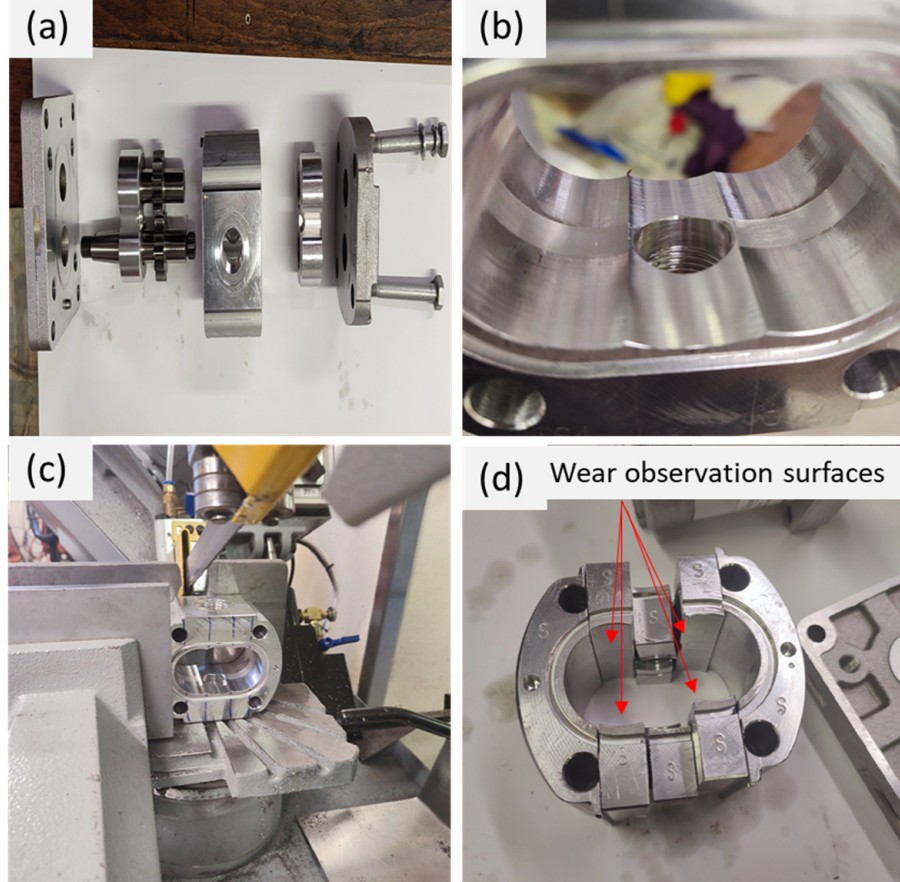

**Figure 4.** (**a**) Disassembled gear pump, (**b**) wear of pump housing, and (**c**,**d**) cutting the pump housing and observing the wear on housing with digital microscope.

### 2.3. Hydraulic Directional 4/3 Valve—Specimen 2

The hydraulic directional 4/3 valve in Figure 5 was used to perform durability tests. The internal leakage of the directional 4/3 valve was measured at ports A and B when the valve was in the zero position, the pressure in the P line was 200 bar, and the T line was continuously relieved. The valve code was DS3-S1/11-N-D24K1 and was intended for controlling the direction of the flow. The valve had static mineral oil-resistant seals, 24 V solenoids controlling the spool in the housing, and standard electrical control connections (DIN 43650).

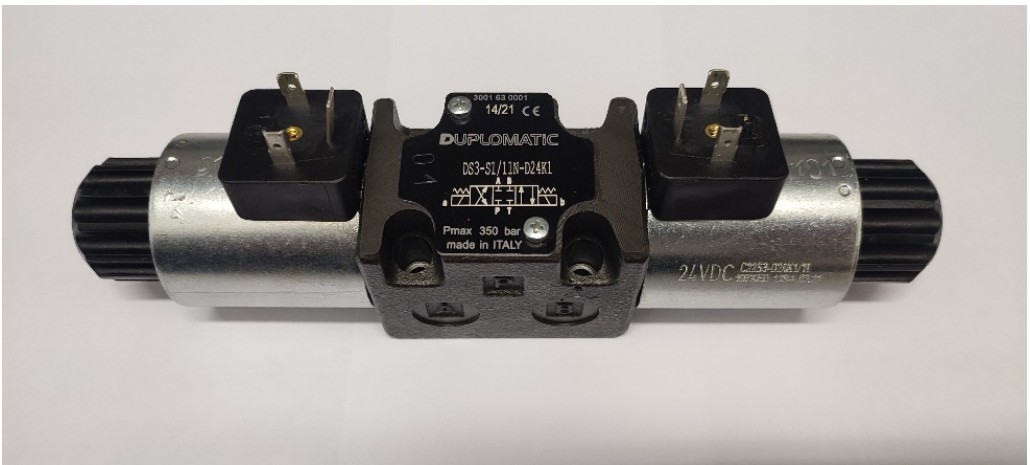

**Figure 5.** Hydraulic directional 4/3 valve DS3-S1/11 N-D24 K1 with A, B, T, and P ports closed in neutral position.

The maximum permissible pressure at the pressure port is 350 bar and is designed for a flow rate of up to 100 L/min. The ambient temperature range in which the valve will operate properly is between −20 °C and 50 °C, and the hydraulic fluid temperature must be between −20 °C and 80 °C for the valve to operate smoothly. The valve body is made of cast iron, and the spool is made of tool steel. These two parts of the whole assembly form a tribological pair, and between them the internal leakage was measured. The manufacturer specifies the pressure drop as a function of the flow through the valve's ports [28].

### 2.4. Hydraulic Motors—Specimen 3

The hydraulic motor manufactured by M+S Hydraulic (Figure 6), type MMFS50C, has a displacement of 50 cm$^3$/rev and a maximum rotation speed of 400 rpm. The maximum continuous torque that can be achieved is 45 Nm, and the maximum output power is 3.2 kW. The maximum pressure difference between inlet and outlet pressures can be 105 bar, and the maximum flow rate can be 25 L/min. The minimum rotation speed is 20 rpm. For the motor to work, the fluid must be mineral oil based. The working temperature of the fluid should be in the range between −40 and +140 °C, and the optimum viscosity of the fluid is between 20 mm$^2$/s and 75 mm$^2$/s. The required cleanliness of the fluid for the operation of the hydraulic motor is 18/16/13 according to the standard ISO 4406.

The pressure loss was through the hydraulic motor when the flow rate changed. At a flow rate of 9 L/min, there were 6 bar losses through the hydraulic motor, while at 23 L/min, there were about 18 bar pressure losses [29]. The hydraulic motor was not loaded during the test. Measurements were made by mounting a rotor with 6 evenly spaced attachment points (60° angular separation) around a rotor on the hydraulic motor. The rotor was fixed at a specific fixing point so that the shaft of the hydraulic motor could not rotate. Then, the internal leakage at each point was measured at a pressure of 100 bar. Excess hydraulic fluid flowed back into the reservoir through the safety valve (pos. 11, Figure 2). The internal leakage at each attachment point was measured.

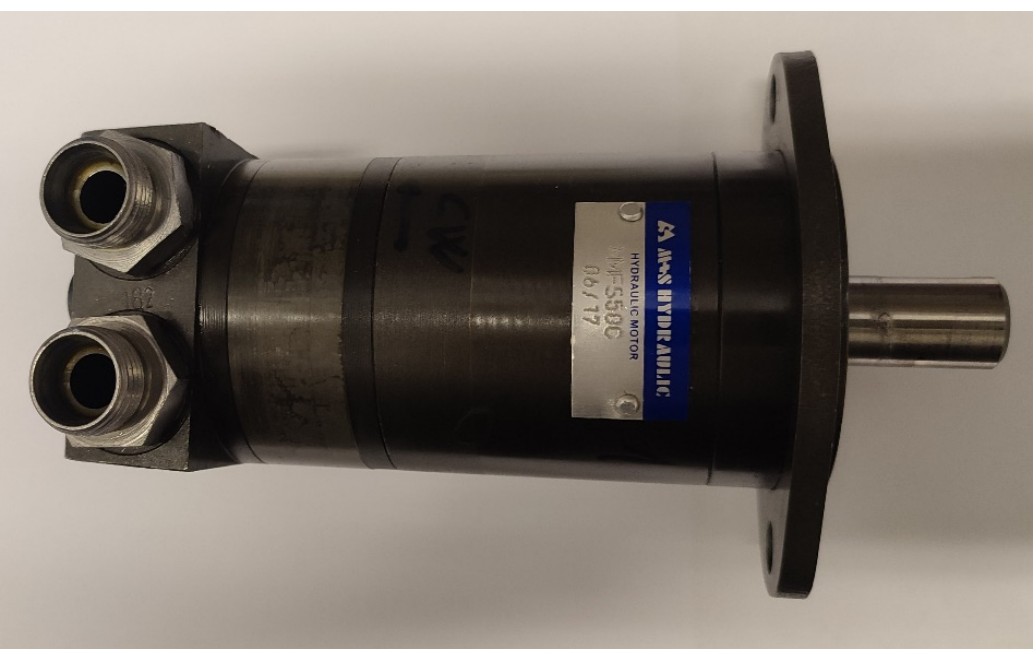

**Figure 6.** Hydraulic motor type MMFS50C.

### 3. Results

The test with the wear particle test lasted 11.27 million cycles (624 h), while the medium test dust test lasted only 1581 cycles (5.3 min), which significantly affected all of the above results.

### 3.1. Cleanliness

The change in cleanliness was a function of the number of cycles performed by the solenoid 4/3 directional valve. The cleanliness was measured before starting the test, after filtration, and during the operation of the test. The lowest value was 16/14/12, according to the standard ISO 4406, and this was after filtration. Figure 7 shows that at the beginning of the test, there were 35,128 particles larger than 2 μm, 14,192 particles larger than 5 μm, 2985 particles larger than 15 μm, 1200 particles larger than 25 μm, 250 particles larger than 50 μm, and 15 particles larger than 100 μm in 100 mL of oil. Then, 5 g of wear particles was added, and cleanliness was measured at 21/21/20. According to this, 1,357,614 particles were larger than 2 μm, 763,850 particles were larger than 15 μm, and 23,725 particles were larger than 100 μm. After 204,000 cycles, the cleanliness stabilized at 21/20/17, so 1 g of wear particles was gradually added every two days (30,000 cycles) to achieve the desired cleanliness level of 22/21/20. The first addition of 1 g of wear particles at 204,000 cycles changed the cleanliness to 22/21/20 and remained unchanged throughout the test, despite the addition of another 8 g of wear particles.

On the test rig contaminated with test dust, the initial cleanliness after filtration was 16/15/13 (48,000 particles were larger than 4 μm, 24,000 particles were larger than 6 μm, and 12,000 particles were larger than 14 μm) and then, 5 g of test dust was added. After that, the cleanliness was 21/21/21 (1,500,000 were larger than 4 μm, 6 μm, and also 14 μm). After 1581 cycles (after 5.3 min), the pump was worn out and the test was terminated (Figure 8).

### 3.2. Gear Pump

3.2.1. Volumetric Efficiency of Gear Pump

At atmospheric pressure (1 bar), the volumetric efficiency of the pump tested with wear particles did not change throughout the operating time. At 100 bar, the maximum volumetric efficiency was noticeably reduced. Even at the beginning, the volumetric efficiency was lower at higher pressures. At 3,620,000 cycles, the pump could not reach

200 bar and dropped to 130 bar. When the pressure was reset to 200 bar, it functioned normally again. At 4,480,000 cycles, the pump started to reduce pressure significantly, so it was no longer possible to maintain the pressure at 200 bar, so the pressure was reduced to 180 bar. After 4,480,000 cycles, the measurements at 200 bar were much shorter (5–10 s) because the pump could no longer keep the pressure as high, so the measurements were less accurate. Up to 4,480,000 cycles, the volumetric efficiency decreased linearly. The efficiency of the pump was 0.83 at the beginning at 100 bar, but only 0.65 at the end of the test. At 150 bar, the efficiency was 0.77 at the beginning and 0.51 at the end of the test. At 200 bar, the efficiency was 0.69 at the beginning of the test and then dropped to 0.38. The efficiency of the pump tested with test dust dropped from 0.87 to 0.11 at 100 bar, and at 150 and 200 bar, it was not possible to take a measurement because the pressure had already dropped, so the efficiency at these pressures was 0 (Figure 9).

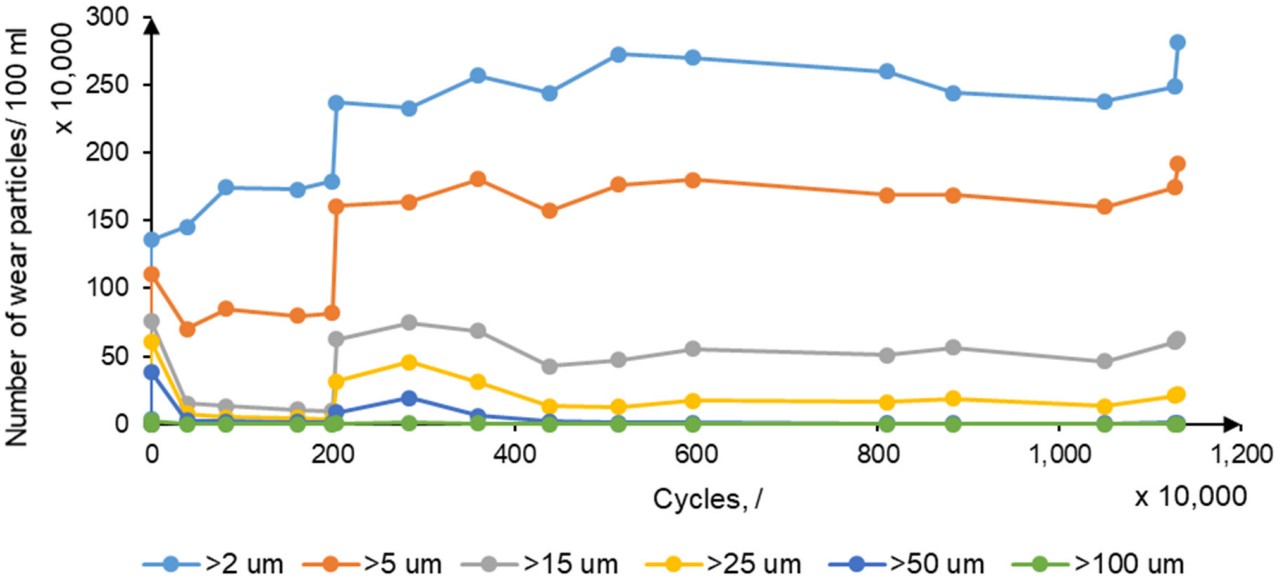

**Figure 7.** Cleanliness level of oil contaminated with wear particles during long-term test.

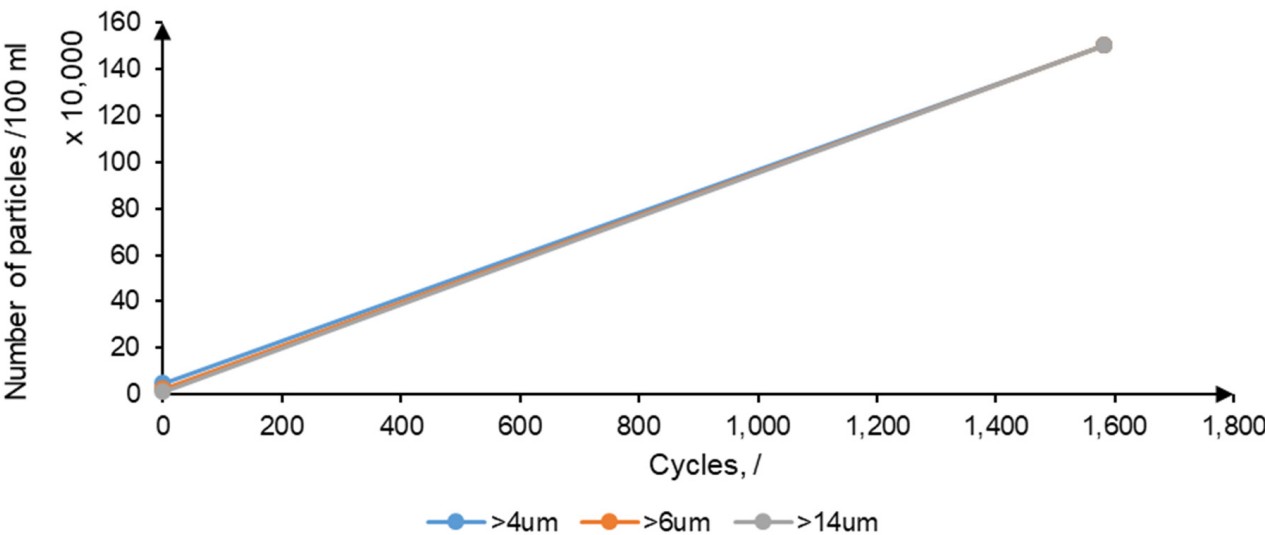

**Figure 8.** Cleanliness level of oil contaminated with medium test dust during long-term test.

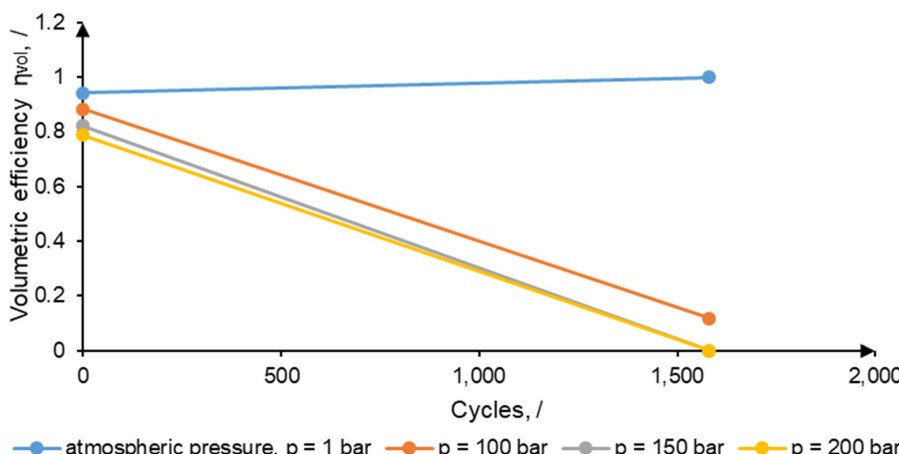

**Figure 9.** Volumetric efficiency of hydraulic gear pump tested with test dust.

The actual efficiency of the pump was not significantly different from the theoretical value. The actual rotational speed of the electric motor shaft and thus of the pump was compared with the theoretical rotational speed of the shaft of the electric motor. The actual rotational speed reduced due to the pressure in the system increases, as can be seen in Table 2. At different pressures, the theoretical motor speed differed from the actual one by a maximum of 1.02%.

3.2.2. Surface Inspection of Hydraulic Gear Pump

Wear on three gear pump housings from a production series was studied. The first was a new pump that was examined. The second pump was tested for 624 h of operation with wear particles. The third pump was tested for 5.3 min with medium test dust. The hardness of the pump housing, which was a type of aluminum alloy, was measured between a minimum of 67 $HV_{0.5}$ and a maximum of 85 $HV_{0.5}$ for all three housings. Internal leakage was present where there were gaps between the pressurized and unpressurized (suction) sections. Wear of the pump housing was observed on the pressurized and suction sides. Figure 10 shows the wear of the housing on the suction side where the gear was in contact with the housing. The width of the worn band was measured as $9.55 \pm 0.05$ mm. The width of the gear that was in contact was $10.00 \pm 0.05$ mm.

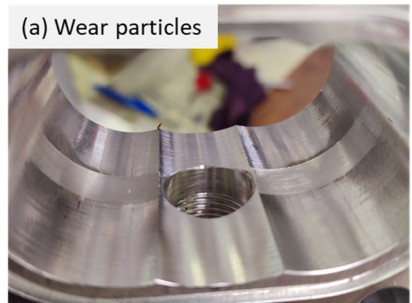
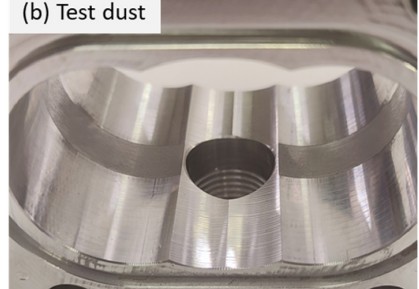

**Figure 10.** Wear of the housing on the suction side of (**a**) pump tested with wear particles and (**b**) pump tested with medium test dust.

The undamaged surface of the pump on the suction side tested with wear particles is shown in Figure 11(a1) and the surface tested with test dust is shown in Figure 11(b1). Figure 11(a1,a2) show the undamaged surface and the boundary between worn and undamaged surfaces of the housing tested with wear particles. There are some horizontal scratches perpendicular to the sliding direction due to wear particles caught between the housing and the side plate (Figure 11(a1,a2)). These particles erode the surface due to the pressure difference. No such horizontal scratches can be seen on the unworn surface of the housing

tested with test dust (Figure 11(b1,b2)) since the test was very short (5.3 min). On the worn surface (Figure 11(a3)), scratches can be seen in the same direction as during sliding, indicating a two-body abrasion where the gear abraded the housing. In Figure 11(b3), there are more scratches in random directions, indicating a three-body abrasion mechanism. The particles that are moving with the fluid are as small, or smaller, as the clearance between the gear and the housing. These are critical size particles, which provide the most damage [18]. Particles also have the ability to get stuck in the softer material (in this case, housing) and then abrade the counter part of the tribological pair, which, in this case, are the gears. Mechanisms such as micro-cutting, micro-ploughing, and micro-rolling occurred.

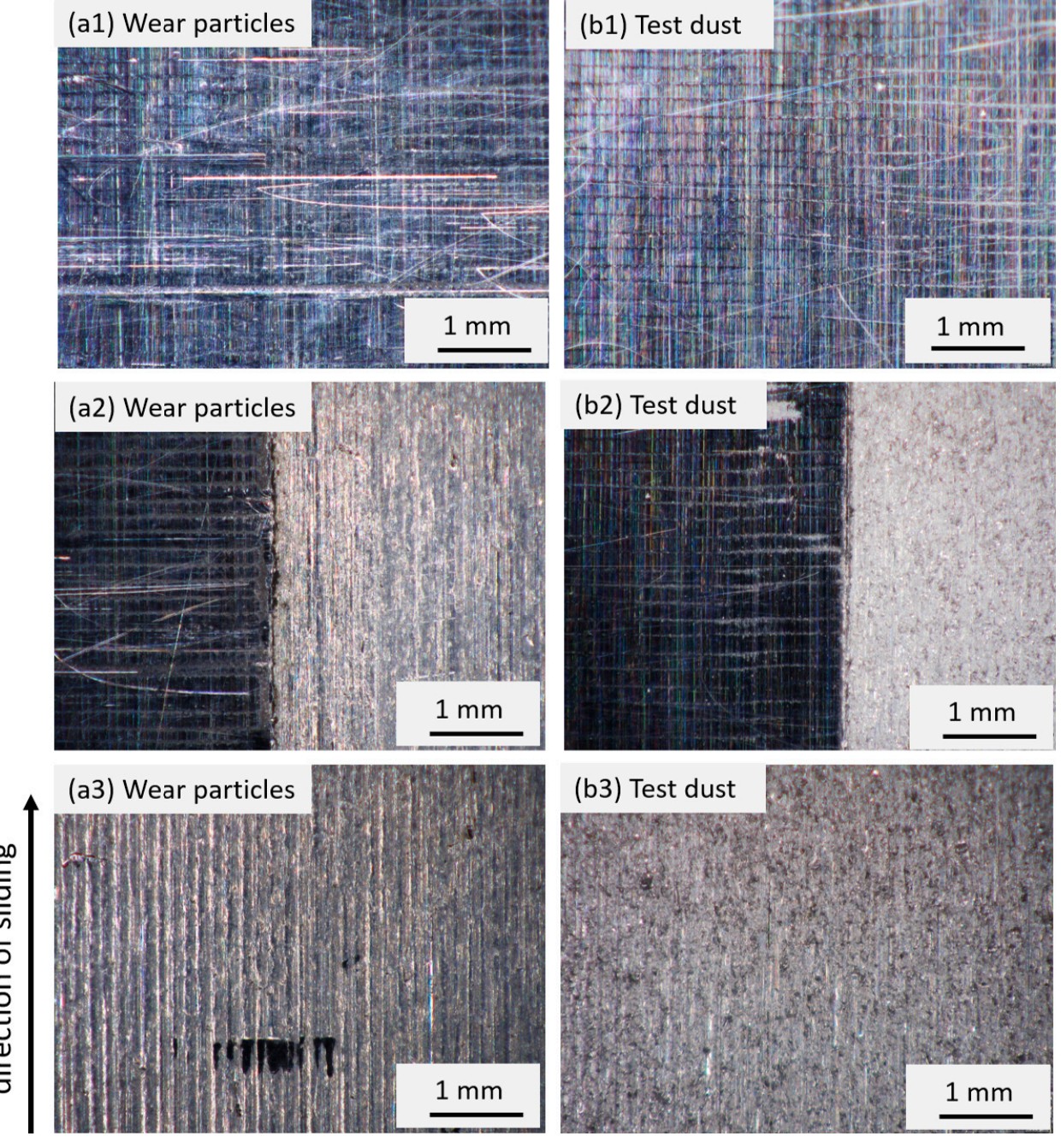

**Figure 11.** Suction side of the pump housing: (**a1**,**b1**) unworn surface, (**a2**,**b2**) border between unworn and worn surfaces, and (**a3**,**b3**) worn surface.

### 3.3. Directional 4/3 Valve

3.3.1. Internal Leakage Measurements on Directional 4/3 Valve

The internal leakage of the directional 4/3 valve was measured at ports A and B when the valve was in the zero position, the pressure at port P was 200 bar, and port T was continuously relieved. The results of internal leakage from port P to port A and from port P to port B are shown in Figure 12. Overall, a linear increase in leakage can be seen. At the beginning, the leakage from port P to port A was 0.02 L/min and from port P to port B was also 0.02 L/min. For the presentation of the results, only the path from port A to port P is described below. At about 8.5 million cycles, the leakage increased to a maximum value of 0.12 L/min, but at the end, the leakage decreased to 0.09 L/min. Not all measurements are consistent with a linear increase in leakage. The reason is the change in position of the control spool in the valve housing. When switching over, the control spool in the valve can rotate uncontrollably, which would lead to deviations from the measurements. The cumulative mass of wear particles is introduced on the secondary axis because additional wear particles during the test were added.

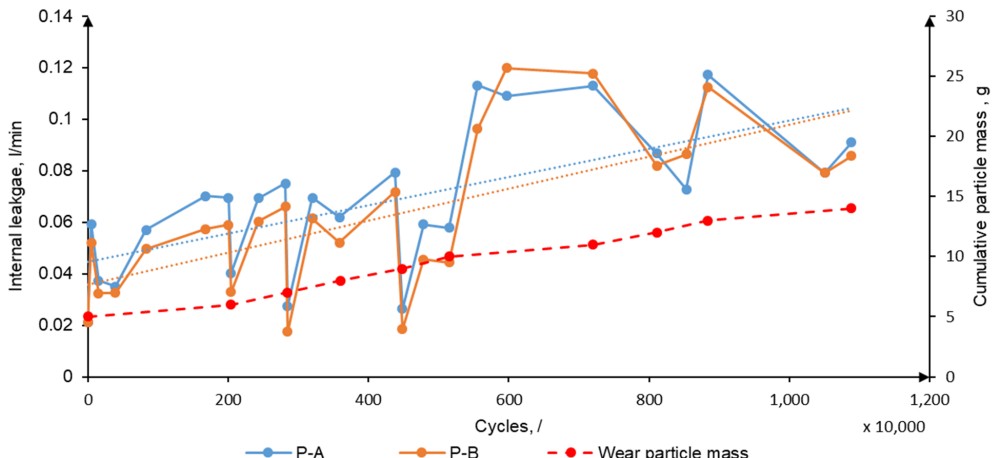

**Figure 12.** Directional 4/3 valve internal leakage measured at ports A and B during durability test with wear particles.

Measurements of the leakage of the directional 4/3 valve tested with test dust were made at the beginning of the test. A first measurement was made at 200 bar. It was 0.05 L/min at port A and 0.04 L/min at port B. At the end of the test, the valve leakage was 0.12 L/min at port A and 0.11 L/min at port B and the leakage increased by a factor of three (Figure 13).

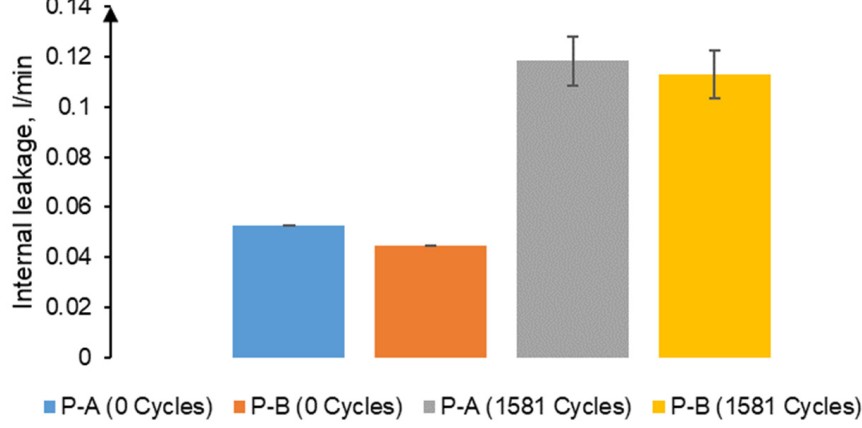

**Figure 13.** Internal leakage of directional 4/3 valve at A and B ports, tested with medium test dust at the beginning of the test and at the end of the test at the additional test rig.

### 3.3.2. Pressure Drop on Directional 4/3 Valve during Operation

Figure 14 shows the pressure conditions on lines A and B at the beginning of the test and at the end of the test (11.27 million cycles) with wear particles over a period of 0.5 s. At the beginning, the system was able to maintain the pressure at 247 bar, but later it was lowered because the system overheated and the safety temperature switch installed in the tank shut down the motor and pump at 70 °C. On site, the temperatures were even higher. The directional 4/3 valve was switched at a frequency of 5 Hz so that the pressure on both line A and line B changed from 247 bar to 1 bar and back to 247 bar on the lines at the beginning of the test. At the end of the test, due to excessive wear on the pump, the system only reached a pressure of 149 bar, as one can see in Figure 14. Thus, the system pressure dropped from the original 247 bar to 149 bar, which corresponds to a pressure drop of 48 bar from the beginning to the end of the operation. Zhang et al. [21] commented on worn sharp edges of spool in an electro-hydraulic spool valve which lead to degradation in leakage flow and degradation in pressure gain (pressure change). At the end of the test, the pressure changes from 149 bar to 1 bar and back to 149 bar with the same frequency of 5 Hz.

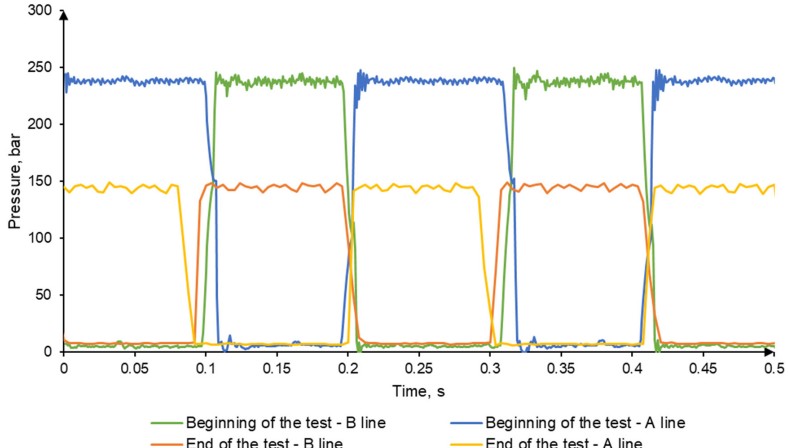

**Figure 14.** Comparison of pressures on A and B lines during operation with wear particles at the beginning and end of the test on pressure ports A and B on directional 4/3 valve.

The pressure conditions on the A and B lines were measured at the beginning of the test and at the end of the test with the test dust over a period of 0.5 s. Figure 15 shows the pressure on the A and B lines, where the system reached 247 bar. At the end of the test (after 5.3 min of operation of the system), pressure in the previously mentioned lines dropped to 27 bar, as it was not possible for the system to reach a higher pressure due to the wear of the pump.

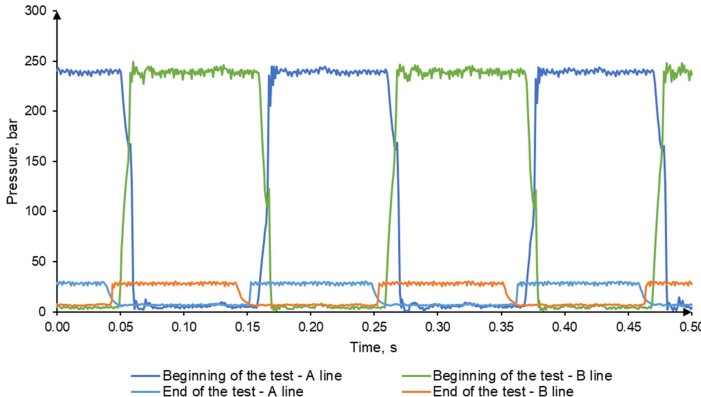

**Figure 15.** Comparison of pressures on A and B lines during operation with test dust at the beginning and end of the test on pressure ports A and B on directional 4/3 valve.

### 3.4. Internal Leakage Measurements on Hydraulic Motor

Figure 16 shows the results of the measurements of the internal leakage of the tested hydraulic motor in attachment point 1. The internal leakage increased slightly during the test with wear particles (Figure 16). The leakage at the beginning of the test was 3.94 L/min and at the end it was 3.49 L/min. Since the hydraulic motor was not loaded and ran freely, the wear particles had little effect on the evolution of wear and thus on the increase in leakage. Leakage was also high due to the design (valve plate, gear, and housing) and operating conditions of the orbital hydraulic motor. Probably, the heat accumulation in the hydraulic motor was so large that the internal leakage increased due to the local heating of the oil.

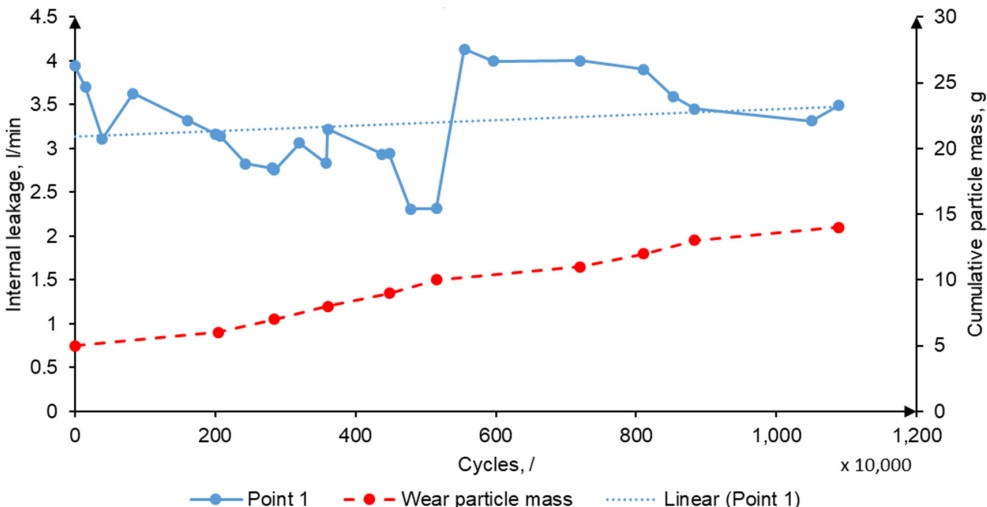

**Figure 16.** Measured internal leakage of a hydraulic motor tested at attachment point 1—test with wear particles.

The internal leakage on the hydraulic motor tested with medium test dust at a pressure of 100 bar was 0.95 L/min at the beginning of the test at attachment point 2 and 0.76 L/min at the end of the test. The following motor reached only 1581 cycles without load, which is a short test for such a hydraulic component; therefore, the leakage did not increase.

## 4. Discussion

### 4.1. Comparison of Hydraulic Gear Pumps Tested with Wear Particles and Test Dust

Initially, a test with wear particles at cleanliness level 22/21/19 and a test at cleanliness level 21/21/21 with MTD was performed. A comparison of the volumetric efficiencies of the pumps shows that the hydraulic system tested with medium test dust failed significantly faster than the hydraulic system tested with wear particles (Figure 17). The flow rates on the far left of Figure 17 show the internal leakage of the pump tested with test dust at four different pressures, and the rest of the figure shows the measured flow rates at four different pressures of the pump tested with wear particles. From the measurements, although the oil cleanliness was better when tested with test dust, the pump wore out much faster. Particles when trapped between two relative moving surfaces cause three-body abrasion which can be further divided into micro-ploughing, micro-rolling, and micro-cutting wear mechanisms [8]. The hardness of particles plays a crucial role because wear particles found in real hydraulic systems have lower hardness than the test dust, which is intended for testing hydraulic components [20–23].

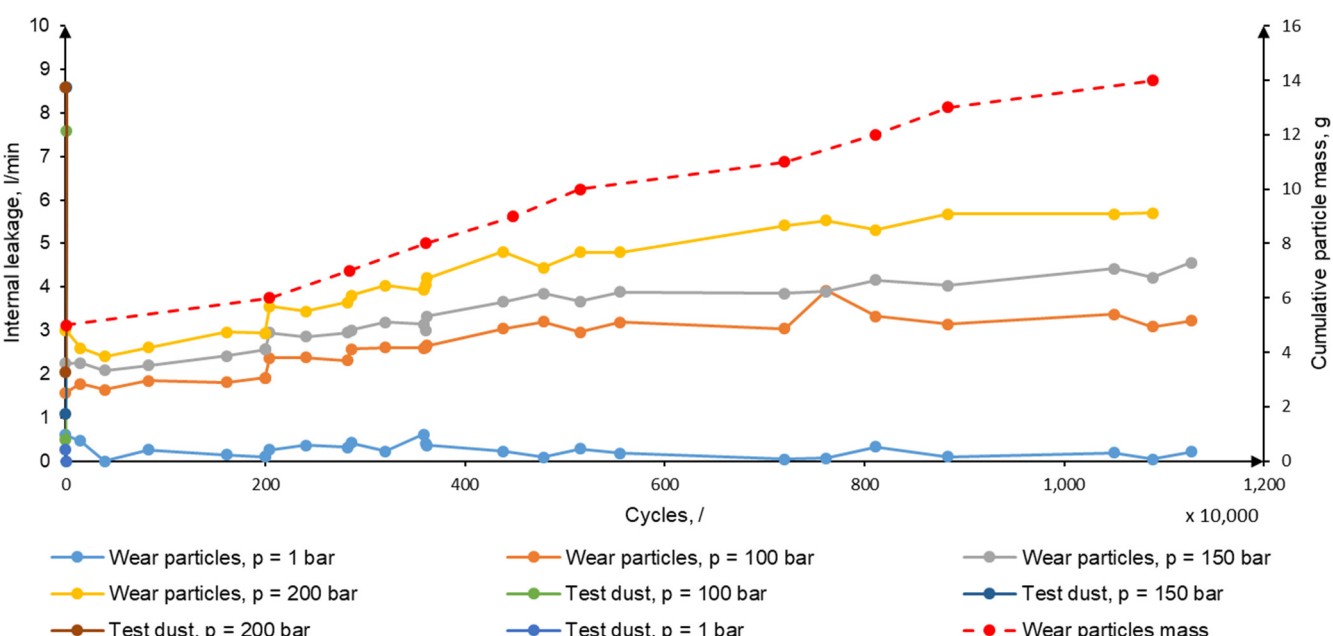

**Figure 17.** Comparison of volumetric efficiencies of hydraulic pump at different operating pressure points (0 bar, 100 bar, 150 bar, and 200 bar), tested with test dust (far left) and wear particles.

Evaluation of the degradation and prediction of hydraulic gear pump life is closely related to leakage characteristics and thus volumetric efficiency. As wear increases, leakage increases, causing volumetric efficiency to decrease [20]. The volumetric efficiency of the pump tested with wear particles was 0.83 at the beginning of the test at 100 bar, but only 0.65 at the end of the test after 624 h of operation. The volumetric efficiency of the pump tested with test dust decreased from 0.87 to 0.11 at 100 bar in only 5.3 min of operation. The reduction in volumetric efficiency at pressure 100 bar was 0.18 for the gear pump tested with wear particles and 0.76 for the gear pump tested with test dust. Wear can be minimized by designing pump leakage to be distributed over a large area. If leakage is confined to small areas, wear can be rapid and the flow degradation rate will be correspondingly high. Designing the pump to distribute leakage over a larger area may result in reduced efficiency. However, this will be compensated by a reduction in the wear rate and, consequently, a longer life [23]. By using more wear-resistant materials, pump housing wear should be reduced.

*4.2. Comparison of Directional 4/3 Valves Tested with Wear Particles and Test Dust*

When comparing the final measurements of internal leakage between the two valves, one tested with wear particles and the other with test dust, the following differences occur. The valve tested with wear particles had a leakage of 0.09 ± 0.01 L/min at port A in the zero position, while the valve tested with dust (also at port A) had a leakage of 0.12 ± 0.01 L/min in the same position at 200 bar. Figure 18 shows the data for ports A and B. It can be seen that the valve tested with wear particles and an oil cleanliness of 22/21/19 completed 11.27 million cycles, while the valve tested with MTD and an oil cleanliness of 21/21/21 completed only 1581 cycles. This is a small amount of leakage, which is common with worn valves in both cases, but the differences are still notable. The volumetric efficiency of the valve tested with wear particles was 0.998 at the beginning of the test and decreased to 0.990, and the volumetric efficiency of the valve tested with test dust was 0.995 and decreased to 0.986. The volumetric efficiency of the valve tested with wear particles decreased by 0.008 and that of the valve tested with test dust decreased by 0.009. The difference in the reduction in volumetric efficiency is much smaller (less than 1%) compared to the pumps. The elements that come into contact with the dust, and are susceptible to wear, are the control spool, made of tool steel, and the valve housing made

of gray cast iron. The mentioned tribological pair is much more wear-resistant than the gear pump housing, which is made of aluminum alloy. Another reason for the difference in wear is that the gear pump has a higher contact pressure between the surfaces of the elements than the directional 4/3 valve.

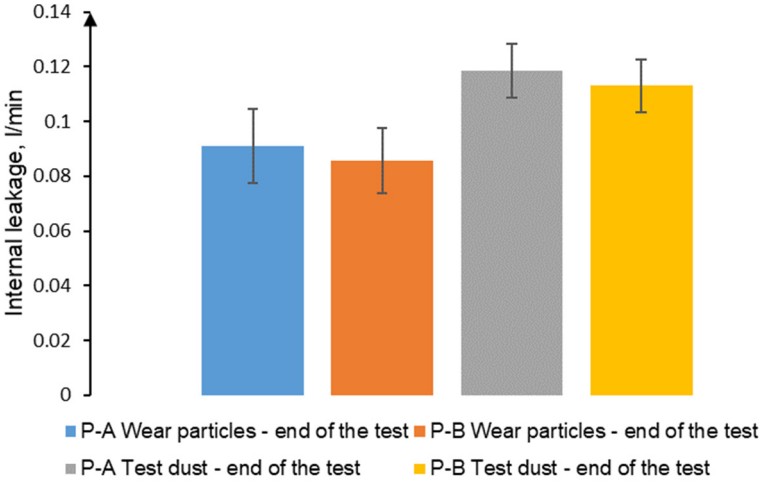

**Figure 18.** Comparison of internal leakage on directional 4/3 valve, tested with wear particles and test dust after the end of the test.

The endurance test of the hydraulic directional 4/3 valve revealed a minor internal leakage, which can be seen in Figure 18, when tested with wear particles and when tested with test dust. It should be pointed out that the valve tested with wear particles was less worn than the one tested with test dust, even though it was in operation for 624 h longer. It can be seen from Figure 12 that the leakage measurements are scattered. One possible explanation for such variation could be the position of the control spool with respect to rotation. If the housing and control spool are in a position where both worn surfaces are in contact, increased leakage may occur.

### 4.3. Wear Parameter

The equations for calculating the leakage through an annular gap (Equation (3)) [30–32] show the dependence of the pressure changes ($\Delta p$) between gap height ($s$), gap diameter ($D_{sr}$), gap length ($L$), viscosity ($v$), and density ($\rho$) of the medium.

$$Q_{\mathrm{L}} = \frac{\pi \cdot \Delta p \cdot D_{sr} \cdot s^3}{12 \cdot \rho \cdot v \cdot L} \tag{3}$$

with time elements in sliding contact are exposed to wear; therefore, the gap, especially the height (s), increases and the leakage increases. If the wear is rapid, the increase in leakage is also rapid. If there is some wear parameter $K$ that includes the slot diameter ($D_{sr}$), the slot length $L$, and the slot height ($s$), then it can be written with Equation (4):

$$K = \frac{Q_{\mathrm{L}} \cdot 12 \cdot \rho \cdot v}{\pi \cdot \Delta p} \tag{4}$$

In this way, the wear parameter can be determined, which should be the same regardless of the pressure at which the internal leakage was measured. In Figure 19, the curves overlap well enough at 100 bar, 150 bar, and 200 bar. Researchers from the University of Tulsa and the University of Petroleum-Beijing studied the degradation and wear of an electric submersible pump (ESP) with gas–liquid–solid flow: experiments and a mechanical model [33]. When the increase in the gap in the water pump with the addition of air and sand is compared to the increase in the gap of gear pump housing, a similar wear trend, as in the case of Figure 19, occurs.

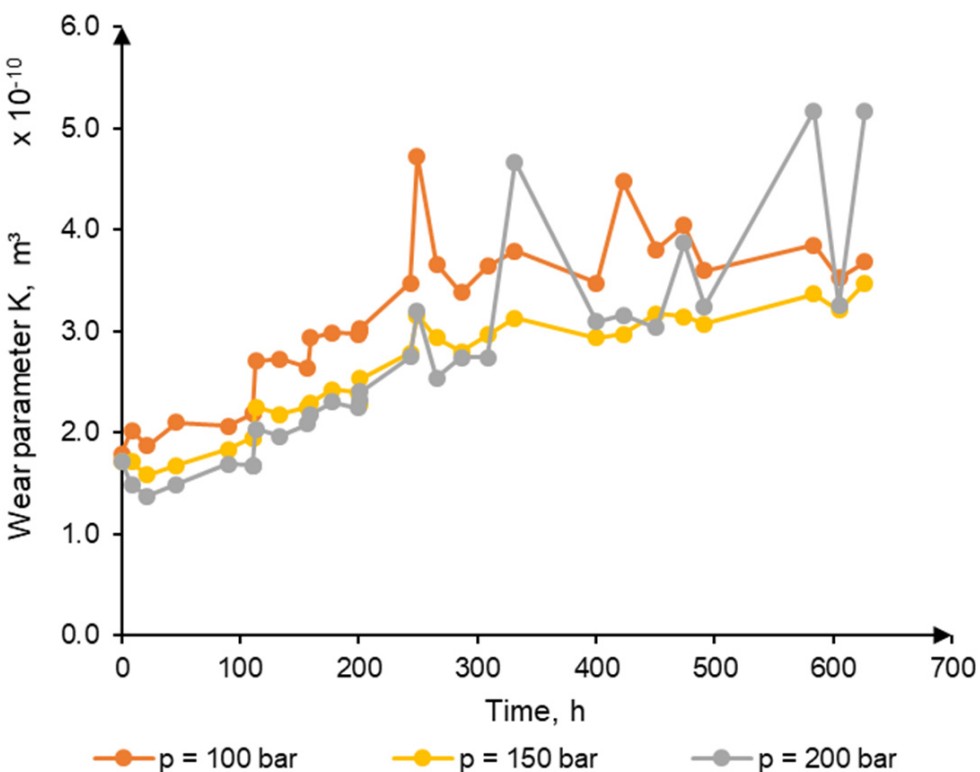

**Figure 19.** Wear parameter *K* depending on the time operation of the pump.

## 5. Conclusions

Two durability tests of hydraulic systems were performed, one with wear particles and the other with test dust. A comparison of volumetric efficiency and leakage measurements between the two tests was presented.

1.  The hydraulic system tested with wear particles failed after 624 h of operation because the pump could not reach a pressure higher than 149 bar, which is insufficient for most hydraulic applications. The hydraulic system tested with test dust failed after 5.3 min of operation because the pump could not reach a pressure higher than 27 bar.

2.  The cleanliness of the oil, including the wear particles, was 22/21/19, and the cleanliness of the oil, including the test dust, was 21/21/21. The test results show that the wear particles are less abrasive than the test dust, which can be attributed to the different operating times of the two hydraulic systems and the relatively equal concentration of contaminants (wear particles and test dust) in the hydraulic oil in each system.

3.  The volumetric efficiency of the pump tested with wear particles decreased from 0.83 to 0.64 after 623 h of operation with a decrease in efficiency of 0.18. For the pump tested with test dust, the efficiency dropped from 0.87 to 0.11 within 5.3 min with a decrease of 0.76.

4.  The volumetric efficiency of the valve tested with wear particles decreased from 0.998 to 0.990, corresponding to a 0.008 decrease in efficiency, and the valve tested with test dust had a decrease from 0.995 to 0.986, corresponding to a 0.009 decrease in efficiency. Compared to the drop in efficiency of the pump, the drop in efficiency of the valve is negligible. The difference is due to the fact that the aluminum housing of the gear pump is a softer material than the cast iron housing of the directional 4/3 valve and therefore wears out much faster. It is also recognized that the valve is tribologically less loaded than the gear pump.

5.  The wear parameter represents the amount of leakage as a function of various pressures and indicates the amount of wear of a particular hydraulic component. For the

pump tested with wear particles, the value is $4 \times 10^{-13}$ m$^3$ at the beginning of the test and varies from $7.5 \times 10^{-13}$ to $1.1 \times 10^{-12}$ m$^3$.

6.  An important part for improving hydraulic systems operation is filtering. Extra care should be taken when new hydraulic is put into the existing operating system in order to avoid contamination, especially with hard particles, such as test dust (SiO$_2$, Al$_2$O$_3$, etc.). New hydraulic fluid should be filtered and the cleanliness of the fluid should be measured, and if the cleanliness is equal or better than the one in the system, only then can fluid be put in the system. Particles cannot be entirely removed from the hydraulic fluid because new particles are also generated due to components wear during the operation of the machine, and furthermore, the smallest particles cannot be removed with the filter because the grid of the filter material is too rough.

7.  A higher concentration (greater number) of particles causes a higher possibility that these particles will damage the sealing surfaces and wear surfaces. With an increase in wear, the gap between the sealing surfaces increases and therefore the leakage increases. Test dust has a higher hardness than wear particles; therefore, it will cause a larger amount of wear than wear particles in the same operation time. With further testing at different fluid cleanliness levels with wear particles and test dust, the acceleration factor can be determined.

*Novelty and Aspects of the Study*

(1)  Two experiments and the comparison between test dust and wear particles were presented for the first time.

(2)  The two identical test rigs and test procedures with identical hydraulic components showed that the test dust has a more damaging effect than the wear particles present in the real hydraulic system.

(3)  The acceleration factor in tests with test dust is about 7000.

(4)  The most wear-prone component is the gear pump due to its aluminum alloy housing.

(5)  The directional control valve was tribologically less loaded than the gear pump, which consequently had a more constant volumetric efficiency.

(6)  Filtration is one of the most important procedures that every hydraulic system should incorporate to provide sufficient fluid cleanliness and prolong the service life of the system.

**Author Contributions:** Conceptualization, N.N. and F.M.; methodology, N.N. and F.M.; validation, N.N., F.M. and A.T.; formal analysis, N.N.; investigation, N.N., A.T. and F.M.; resources, F.M.; data curation, N.N.; writing—original draft preparation, N.N.; writing—review and editing, N.N., A.T., M.K. and F.M.; visualization, F.M. and A.T.; supervision, F.M.; project administration, M.K.; funding acquisition, F.M. and M.K. All authors have read and agreed to the published version of the manuscript.

**Funding:** This research has been financially supported by Slovenian Research Agency as part of the research program no. P2-0231 with grant project numbers L2-4474 and L2-2618.

**Institutional Review Board Statement:** Not applicable.

**Informed Consent Statement:** Not applicable.

**Data Availability Statement:** All data, models, and code generated or used during the study are available from the corresponding author by request.

**Acknowledgments:** Authors would like to acknowledge the financial support of the Slovenian Research Agency ARRS in the program group P2-0231 of an area (2.11.03), Rok Jelovčan for his support in providing help in the construction and assembly of the hydraulic test rig, and Franci Kopač for measuring the hardness of the specimens.

**Conflicts of Interest:** The authors declare no conflict of interest.

## Glossary

| | |
|---|---|
| **Wear particles** | Particles generated due to wear between the sliding surfaces of the elements of a component. |
| **Service life** | Product or component period can continuously perform its intended function. |
| **Hydraulic fluids** | Hydraulic fluid is the medium used to transmit power in hydraulic machines. |
| **Filtration** | Separating substances based on their physical and chemical properties. Usually involves the removal of solid particles from a mixture containing both solids and liquids. |
| **Cleanliness of fluids** | According to the standard ISO 4406, cleanliness is the particle concentration and size distribution in a fluid and is evaluated with three size categories: number of particles greater than 4 µm, particles greater than 6 µm, and particles greater than 14 µm. |
| **Acceleration testing** | Process of testing a product or component by subjecting it to conditions beyond normal operating parameters in order to rapidly uncover faults and potential causes of failure. |
| **Gear pumps** | Pumps use the meshing of gears to pump fluid by displacement. They are one of the most common types of pumps for hydraulic fluid power applications. |
| **Hydraulic components** | Part of a hydraulic system that performs one or more of the following functions: generation, transport, or control of hydraulic power, actuation, and fluid maintenance. This includes pumps, pipes, valves, cylinders, hydraulic motors, filters, tanks, etc. |
| **Internal leakage** | Inside components, there is a gap between the two elements with relative velocities. When there is a pressure difference, a certain flow escapes through this gap, which is called internal leakage. Internal leakage is inevitable in components that use elements with sliding contacts. As the sliding surfaces wear, the internal leakage increases. |
| **Directional valve** | Hydraulic component that controls the direction of flow of hydraulic energy and thus fluid. |
| **Performance monitoring** | A set of measurements and tools that you can use to determine how well the applications are working and how efficient the system is. |
| **Troubleshooting** | A systematic approach to problem solving designed to track down the problem with appropriate measurement equipment and correct the problem by replacing or repairing the system component. |

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
