# Peer review of "Degradation of Hydraulic System due to Wear Particles or Medium Test Dust"

_applsci, doi:10.3390/app13137777_

Round 1

Reviewer 1 Report

Dear authors, the scientific significance of the work is not clear in the article.

This work is a test report. What new aspects in science did she reveal?

What advice should readers get from this work?

Please develop the work towards scientific results that expand knowledge about the subject of research. Write at least some recommendations for improving hydraulic systems.

Author Response

Dear reviewer 1,

thank you for your review. Please find the attached Word file below with responses to your questions. For any further information or questions regarding this paper I hope we stay in touch.

Best Regards,

Nejc Novak

Reviewer 2 Report

1) The entire document has several spacing mistakes. Please check the manuscript, WORD by WORD, LETTER by LETTER.

2) Besides, a COMPLETE FORMATTING CHECK is required.

3) Explain/describe/define the following terminologies in the manuscript, in a separate GLOSSARY section (VERY IMPORTANT):

- Wear particles

- Service life

- Hydraulic fluids

- Filtration

- Cleanliness of fluids

- Acceleration testing

- Gear pumps

- Hydraulic components

- Internal leakage

- Directional value

- Performance monitoring

- Troubleshooting

Also, make this as a nice glossary, and add more terminologies to the paper so that high school, BS and MS students can read, understand and follow the paper clearly.

4) Write the NOVELTY of this work in 5 sentences.

5) Provide details of all the equipment, instruments, and include the model number, country details, etc in the materials and methods section.

6) How were the conditions of tank temperature, and orifice diameter determined?

7) How did the authors set the adjustable pressure differential values and conditions?

8)  50 cm3/rev ..... write 3 in superscript. 

9) Check the entire document for all such formatting mistakes.

10) mm2/s ...............2 in superscript, please.

11) Discuss the following aspects in detail with REF support:

- Cleanliness stabilization

- Worn out of the pump and how could it be resolved

- Volumetric efficiency decrease

- Pressure drop changes

- Role of the theoretical rotational speed of the shaft

- Hardness of the pump housing

- Phenomenon of corrosion

- Trapping of particles between the gear and the housing

-  Control spool functions

- Explain the phenomenon of the damaging effects

12) Please make sure that the graphs are prepared as follows:

- X and Y axis lines are black in colour and also the numbers and text in the axis are in black colour. Do not use GREY colour. Please check the settings of your excel file or software

- Error bars should be provided wherever applicable

- Units are written in a consistent format (check also the text and tables)

- Spacing errors are avoided 

- Delete the outer border in the figure (box type external border)

- Use Arial font in the x and y axis

- In the legends, there should not be any formatting mistakes or spacing mistakes

- Please make sure that there are no grid lines

- Major tick marks should be outside (check your excel file settings and formatting)

- Avoid all the unwanted ZEROs after the decimal point

- Use decimal point and not a COMMA

13) The comments from the reviewers and the editors of APPLIED SCIENCES should be INCORPORATED in the revised manuscript. Therefore, please read the decision letter till the very end.

14) Do not leave any comments unanswered. Also, the author's response file should be DETAILED with the author's response, i.e. it has to show all the new text added, the new changes made to the graphs, tables, etc. Please don't write - DONE, OK, Thanks, Modified, See page XX, we answered it, we implemented it, etc.  

15) ONCE AGAIN = All the comments should be INCORPORATED.

16) The MS should be re-reviewed.

Minor formatting and English check is required

Author Response

Dear reviewer 2,

thank you for your review. Please find the attached Word file below with responses to your questions. For any further information or questions regarding this paper I hope we stay in touch.

Best Regards,

Nejc Novak

Reviewer 3 Report

This work invesitegated the degradation caused by solid particle contamination of hydraulic components experimentally and presented the technique to improve the quality of hydraulic filters. This work is meanful in real applications. Before its acceptance, I have  following comments

(1) On Page 6, Eq. (1) and (2) are same.

(2) On page 12, Figure 18 shoule be Figure 13.

(3) Line 191 on Page 8, it should be Fig.8 shows ...instead of in the Fig.8 shows ...

(4) there are two (3) on Page 18.

(5) if the results will be affected by the number of particles or test dust? How?

(6) If the leakage is affected by age or something else?

some errors should be corrected.

Author Response

Dear reviewer 3,

thank you for your review. Please find the attached Word file below with responses to your questions. For any further information or questions regarding this paper I hope we stay in touch.

Best Regards,

Nejc Novak
